# Towards Interpretable Classification of Leukocytes based on Deep Learning

Stefan Röhrl [* 1]   Johannes Groll [* 1]   Manuel Lengl [1]   Simon Schumann [1]   Christian Klenk [2]   Dominik Heim [2]
Martin Knopp [2 1]   Oliver Hayden [2]   Klaus Diepold [1]

## Abstract

Label-free approaches are attractive in cytological imaging due to their flexibility and cost efficiency. They are supported by machine learning methods, which, despite the lack of labeling and the associated lower contrast, can classify cells with high accuracy where the human observer has a little chance to discriminate cells. In order to better integrate these workflows into the clinical decision making process, this work investigates the calibration of confidence estimation for the automated classification of leukocytes. In addition, different visual explanation approaches are compared, which should bring machine decision making closer to professional healthcare applications. Furthermore, we were able to identify general detection patterns in neural networks and demonstrate the utility of the presented approaches in different scenarios of blood cell analysis.

## 1. Introduction

The complexity of deep learning models is growing in various fields. Medical imaging and clinical decision making are not exempt from this development (Holzinger et al., 2019; Guo et al., 2017). In recent years, deep learning has helped to support and automate many diagnoses, if it didn't even make them possible in the first place (Shen et al., 2017; Lundervold & Lundervold, 2019). Despite the potential benefits, doctors and patients remain skeptical about basing diagnosis and treatment on the output of black box models. Adversarial patterns and malfunctions could easily harm human life in these safety critical scenarios (Zeiler & Fergus, 2014; Rudin, 2019). Recently, the US Food & Drug Administration (FDA) approved several machine learning approaches in medical applications (Benjamens et al., 2020) but adoption could still be faster. From a scientific and regulatory perspective, the developers of such tools are particularly challenged to better address their target groups and to transparently communicate the performance as well as the limitations of their products (Holzinger et al., 2019; High-Level Expert Group on Artificial Intelligence, 2019; Rudin, 2019). In addition, the applications often lack appropriate customization for typical clinical workflows. Decisions are never made without sound evidence, and equipment must meet strict quality specifications. The target group is accustomed to particular types of visualizations that new technologies must adopt to have a chance of gaining trust (Evagorou et al., 2015; Vellido, 2020).

Quantitative Phase Imaging (QPI) is one of these new platform technologies that benefit greatly from the advances in computer vision and machine learning (Nguyen et al., 2022). Microscopes based on QPI are able to capture the optical height of cells without time consuming and costly fluorescence staining. Hence, many hematological (Go et al., 2018; Ozaki et al., 2019) and oncological (Nguyen et al., 2017; Ugele et al., 2018; Paidi et al., 2021) applications were demonstrated in this field. However, the resulting images are widely unknown to biomedical researchers and practitioners as they show only limited resemblance to light microscopy stained images (see Figure 1). Moreover, none of the previous publications put much emphasis on **visually explaining** the results of machine learning models or demonstrating the robustness of the approaches to perturbations.

In this work, we aim to transfer leukocyte classification as one of the most widely used laboratory tests (Horton et al., 2018) from molecular hematology to QPI and deep learning. To do this, we focus on rather small architectures like the *AlexNet* (Krizhevsky et al., 2012) and the *LeNet5* (LeCun et al., 1998), as the cell images do not require thousands of highly specialized filters, and even larger models would contradict our quest for transparency. In the following sections, we will provide a baseline for **differentiating four subtypes of leukocytes** with the proposed network architectures. Regarding **confidence estimation**, we will introduce modifications for variational inference and compare them to the frequentist approach. The predictions of the confidence calibrated models will then be used to test different **visual explanation tools** to support and communicate their decisions to the medical target group. Furthermore, we apply meta-aggregations to derive **general detection patterns** for the distinct cell classes dependent on their confidence

---

[*]Equal contribution  [1]Chair of Data Processing, Technical University of Munich, Germany  [2]Heinz-Nixdorf Chair of Biomedical Electronics, Technical University of Munich, Germany. Correspondence to: Stefan Röhrl <stefan.roehrl@tum.de>.

*Workshop on Interpretable ML in Healthcare at International Conference on Machine Learning (ICML)*, Honolulu, Hawaii, USA. 2023. Copyright 2023 by the author(s).

level. The more the network uses visual properties of the cell that are also important for human experts, the easier it becomes to justify the decisions. Finally, we will apply our findings to common obstacles in the cell analysis workflow and demonstrate the robustness and explainability of the architectures studied.

## 2. Background and Related Work

### 2.1. Confidence Estimation and Calibration

As in many safety critical scenarios, the safe use of clinical decision support systems (CDSSs) can only be ensured if the reliability and the limitations of the model can be accurately stated (High-Level Expert Group on Artificial Intelligence, 2019). Predictions with a low confidence level have to be checked by human experts, e.g. physicians, whereas the CDSS gets more autonomy in cases of high confidence. This approach borrows closely from human decision making, where trust is an important dimension of human interaction. Therefore, considering confidence estimations helps in interpreting predictions of deep learning algorithms and supports the development of a trustworthy interaction of a user with a CDSS (Guo et al., 2017).

A classification model is said to be *calibrated* if the prediction probability is equal to the actual probability of being correct. This behavior can be evaluated using a **reliability plot** (DeGroot & Fienberg, 1983; Niculescu-Mizil & Caruana, 2005), in which the accuracy of a model is plotted as a function of reliability. A perfectly calibrated model is represented as the identity function. For example, Guo et al. (2017) studied the confidence calibration of modern neural networks. While smaller neural networks, such as those proposed in LeCun et al. (1998) or Niculescu-Mizil & Caruana (2005), appear to produce well-calibrated confidence estimates, this is not true for more complex model architectures. Larger model architectures such as *AlexNet* (Krizhevsky et al., 2012) or *ResNet* (He et al., 2016) achieve better performance, they also tend to produce significantly higher confidence values compared to the achieved accuracy.

To counteract this behavior, several methods have been proposed for re-calibrating a model's confidence estimates in post-processing (Platt, 1999; Zadrozny & Elkan, 2002; Naeini et al., 2015; Kull et al., 2019). **Temperature scaling** has been shown to be effective for multiclass ($K > 2$) classification tasks. Here, the network logits $\mathbf{z}_i$ for the $i$-th sample for each class $k \in \{1, ..., K\}$ are scaled by a learned scalar parameter $T > 0$ before entering the softmax function

$$\sigma_{SM}\left(\frac{\mathbf{z}_i}{T}\right)^{(k)} = \frac{\exp\left(z_i^{(k)}/T\right)}{\sum_{j=1}^{K} \exp\left(z_i^{(j)}/T\right)}. \qquad (1)$$

Once the network is trained, $T$ can be optimized based on the validation set. The scaling factor $T$ does not affect the maximum of the softmax function and has in turn no negative impact on the model performance (Guo et al., 2017).

### 2.2. Visual Explanation of Deep Learning Models

In addition to calculating accurate confidence estimates, this work aims to improve the transparency of model predictions by providing visual explanations similar to Ghosal & Shah (2021) or Huang et al. (2021).

**Model-agnostic** methods impose no restrictions on the architecture or training of a model and are therefore flexible in their application. Furthermore, they do not affect the model performance while still offering intuitive explanations even for uninterpretable features of a black-box (Ribeiro et al., 2016b). The popular framework for Local Interpretable Model-agnostic Explanations (LIME) by Ribeiro et al. (2016a) approximates the local behavior of any machine learning model for a given input sample. For this, the interpretable representation of a sample $x \in \mathbf{R}^d$ is modeled as a binary vector $x' \in \{0, 1\}^{d'}$ indicating the presence or absence of important features. For image classification, it is beneficial to apply this representation to contiguous patches, so-called *super-pixels*. Hence, the method is strongly dependent on the chosen segmentation algorithm like *Quickshift* (Vedaldi & Soatto, 2008), *SLIC* (Achanta et al., 2012), or *compact watershed* segmentation (Neubert & Protzel, 2014). The local behavior of the non-linear model $f : \mathbb{R}^d \to \mathbb{R}$ is approximated by a surrogate model $g : \{0, 1\}^{d'} \to \mathbb{R}$ in the linear form of $g(z') = w_g \cdot z'$, with $z'$ being sampled from the neighborhood of $x$. An adaptation for LIME to work with Bayesian predictive models and approximate both mean and variance of an explanation from the underlying probabilistic model is given by Peltola (2018).

**Propagation-based** approaches, in contrast, use the internal structure of a neural network to determine the relevance of features to the model's internal decision-making. *Class Activation Mapping* (Zhou et al., 2016) demonstrated a way to retroactively add location information to a prediction, even though the convolutional layers solely acted as pattern detectors. Generalizing this approach to networks that additionally contain fully-connected layers, Selvaraju et al. (2017) proposed *Gradient-weighted Class Activation Mapping* (Grad-CAM). Another approach to extract information from the network's internals follows the principles of *Backpropagation*. This mechanism is commonly applied to train neural networks and trace back the output weights of a model to the actual feature map (Springenberg et al., 2015). Thus, gradient information that contributes to the prediction of a particular class, i.e., gradients with a positive sign, are propagated through the network and displayed as an explanation. *Guided Backpropagation* combines this gradient information with Grad-CAM to weight these potentially noisy explanations (Selvaraju et al., 2017).

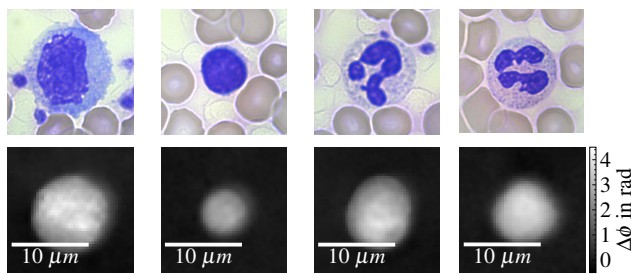

(a) Monocyte (b) Lymphocyte (c) Neutrophil (d) Eosinophil

*Figure 1.* Subtypes of leukocytes: The upper row shows the cells under a light microscope with Giemsa staining (Barcia, 2007) on substrate. The lower row contains the corresponding phase images in suspension using monochromatic light at $\lambda$=528nm.

**Meta-explanations** are methods for aggregating individual explanations to extract general patterns and to draw conclusions about the model's overall behavior. This can be done by clustering the explanations (Lapuschkin et al., 2019), perform a layer-wise *relevance propagation* (Bach et al., 2015), or use *concept activation vectors* (Kim et al., 2018).

## 3. Methodology and Data Acquisition

### 3.1. Quantitative Phase Images of Leukocytes

The data used in this work was captured with a QPI microscope as used by Ugele et al. (2018) and Klenk et al. (2019). The liquid sample stream is focused by a microfluidics chip, allowing tens of thousands of cells to be imaged under near *in-vivo* conditions in a matter of minutes. The resulting phase images are 512×382 pixels in size, each containing multiple leukocytes. Background and noise subtraction is then performed to prepare the images for threshold segmentation and to separate the individual cells into single cell image patches. The entire preprocessing pipeline is described in Appendix A.1. Filtering out debris and defocused cells by requesting a *diameter* $\geq$ 4µm and a *circularity* $\geq$ 0.85 (see Appendix A.2), we obtained a set of $N$=11,008 leukocytes, balanced by their class label. They were randomly split into a training (70%), validation (20%) and test set (10%). Note that *Basophil* cells were excluded from the widely known *Five-Part Differential* data set, as it was not possible to prepare a sufficient number of cells, due to their natural sparsity and our limited number of healthy donors. Consequently, the data set consists of **Monocytes**, **Lymphocytes**, **Neutrophil** and **Eosinophil** cells, forming a *Four-Part Differential*. Typical examples of cell images are shown in Figure 1.

### 3.2. Experimental Setup and Metrics

In this work, we compare the performance of the larger *AlexNet* to the smaller *LeNet5* in the aforementioned four-part leukocyte differential. Thus, the last fully-connected layer was adapted to the four classes. As dropout layers are

necessary to implement variational inference (VI), we introduced one after each fully-connected layer. The single cell patches of 50×50 pixels were scaled to the expected input dimensions. In case of AlexNet, the gray-scale phase images were replicated to three channels. For training, we used ADAM optimization (Kingma & Ba, 2015) with a cross-entropy loss function for $N$ samples and $K = 4$ classes

$$\mathcal{L}_{CE} = -\frac{1}{N} \sum_{i=1}^{N} \sum_{j=1}^{K} y_{i,j} \log(p_{i,j}), \qquad (2)$$

where $y_{i,j}$ is a binary indicator for a correct classification and $p_{i,j}$ is the prediction probability for an observation $i$ of class $j$. The networks' classification performance is assessed using the measures of precision and recall

$$\text{Precision} = \frac{T_p}{T_p + F_p}, \quad \text{Recall} = \frac{T_p}{T_p + F_n} \qquad (3)$$

as well as their harmonic mean

$$F_1 = 2 \cdot \frac{\text{Precision} \cdot \text{Recall}}{\text{Precision} + \text{Recall}}. \qquad (4)$$

Predictions are gathered using *frequentist* deterministic forward-pushes, in the conventional case. The confidence estimation is derived from the softmax output. For variational inference, the dropout layers stay active during testing, resulting in a probabilistic behavior for a single input. These outputs are summarized as *mean*, *median* and *standard deviation* to form a prediction, and the values of 100 independent predictions to form the confidence score.

Besides the reliability plots described in Section 2.1, this work follows Naeini et al. (2015) for evaluating the confidence estimations. Clustering the described confidence estimations in $M = 10$ equally-spaced bins, we are able to estimate the expected calibration error

$$\text{ECE} = \sum_{m=1}^{M} \frac{|B_m|}{n} \left| \text{acc}(B_m) - \text{conf}(B_m) \right| \qquad (5)$$

as a term describing the average confidence/accuracy deviation of each bin $B_m$ weighted by the number of contributing samples. To provide a lower quality bound, the maximum calibration error

$$\text{MCE} = \max_{m=1}^{M} \left| \text{acc}(B_m) - \text{conf}(B_m) \right| \qquad (6)$$

was calculated analogously. Investigating the reliability of the demonstrated approaches, we conducted every experiment in 15 evaluation runs containing independent initializations and data splits.

## 4. Experiments

### 4.1. Model Performance with Variational Inference

As a baseline for the succeeding experiments, we compare our two model architectures in a frequentist and variational

| | Dropout | VI | Metric | Precision | Recall | $F_1$ | Accuracy |
|---|---|---|---|---|---|---|---|
| **LeNet5** | p=0.00 | ✗ | – | 0.922 (1.0e-2) | 0.923 (9.2e-3) | 0.922 (9.4e-3) | 0.927 (9.0e-3) |
| | p=0.25 | ✗ | – | 0.924 (9.9e-3) | 0.926 (1.1e-2) | **0.925** (1.0e-2) | 0.930 (9.1e-3) |
| | p=0.50 | ✗ | – | 0.910 (1.0e-2) | 0.913 (1.0e-2) | 0.911 (1.0e-2) | 0.917 (9.4e-3) |
| | p=0.25 | ✓ | mean | 0.925 (8.4e-3) | 0.927 (9.7e-3) | **0.926** (8.8e-3) | 0.931 (7.6e-3) |
| | p=0.50 | ✓ | mean | 0.910 (1.1e-2) | 0.916 (9.8e-3) | 0.913 (1.0e-2) | 0.918 (9.8e-3) |
| | p=0.25 | ✓ | median | 0.925 (9.1e-3) | 0.926 (1.1e-2) | **0.924** (1.0e-2) | 0.930 (8.8e-3) |
| | p=0.50 | ✓ | median | 0.909 (1.1e-2) | 0.915 (1.0e-2) | 0.911 (1.0e-2) | 0.917 (9.4e-3) |
| **AlexNet** | p=0.00 | ✗ | – | 0.965 (5.2e-3) | 0.962 (4.8e-3) | **0.963** (4.9e-3) | 0.967 (4.1e-3) |
| | p=0.25 | ✗ | – | 0.963 (5.2e-3) | 0.960 (6.1e-3) | 0.962 (5.5e-3) | 0.966 (4.4e-3) |
| | p=0.50 | ✗ | – | 0.963 (6.8e-3) | 0.959 (5.9e-2) | 0.961 (6.3e-3) | 0.965 (5.9e-3) |
| | p=0.25 | ✓ | mean | 0.963 (5.2e-3) | 0.960 (6.0e-3) | **0.962** (5.5e-3) | 0.966 (4.4e-3) |
| | p=0.50 | ✓ | mean | 0.963 (6.8e-3) | 0.959 (5.9e-3) | 0.961 (6.3e-3) | 0.965 (5.9e-3) |
| | p=0.25 | ✓ | median | 0.963 (5.2e-3) | 0.960 (6.1e-3) | **0.962** (5.5e-3) | 0.965 (4.4e-3) |
| | p=0.50 | ✓ | median | 0.963 (6.8e-3) | 0.959 (5.9e-3) | 0.961 (6.3e-3) | 0.966 (5.9e-3) |

*Table 1.* Classification results for the test set over 15 runs using frequentist (VI=✗) and variational inference (VI=✓). The table shows the averaged results. Standard deviation is stated in brackets.

inference setting. To consider both, precision and recall characteristics of the tested models, the $F_1$-score was used as key performance metric. In the case of a frequentist model, the model output was normalized using a softmax function and considered as the prediction value. The prediction values of the probabilistic models were calculated as the mean or median value of 100 independent forward pushes for each sample. Table 1 lists the performance on the test set of 15 independent runs. All model and training configurations showed convergence. In the frequentist setting the AlexNet ($F_1$=96.3%) reaches a slightly better performance than the LeNet5 ($F_1$=92.5%) and featured less variance. The impact of dropout regularization was rather low. For the LeNet5 a moderate dropout rate of $p$=0.25 even improved the classification performance. Hence, in further experiments we use a dropout rate $p$=0.50 for AlexNet and $p$=0.25 for LeNet5 architectures, if not stated differently.

### 4.2. Confidence Calibration

For qualitative analysis, we use reliability plots, which show the accuracy of a model as a function of a confidence score. To this end, the predictions were grouped into $M$=10 equal bins based on their respective confidence estimation. In case of a perfectly calibrated model, the empirical frequency should be an identity of the probability, as indicated with a red line in the following plots. If the frequency for a bin is below this line, the predictions are less accurate than the estimated confidence and the model becomes overconfident. We noticed that the frequencies show a high variance at lower probabilities and thus extended the vanilla reliability plots to box plots in the following figures. In the **frequentist** setting, Figure 2 reveals more stability and a better default calibration of the LeNet5. The larger AlexNet, in contrast, exposes unstable behavior and overconfidence. Applying temperature scaling to both of the models provided a more reliable estimate. The overconfidence reduces tremendously and especially the stability of AlexNet improves. Table 2 registers the effect of the calibration on the ECE and MCE, which in case can be improved by up to 53.8%.

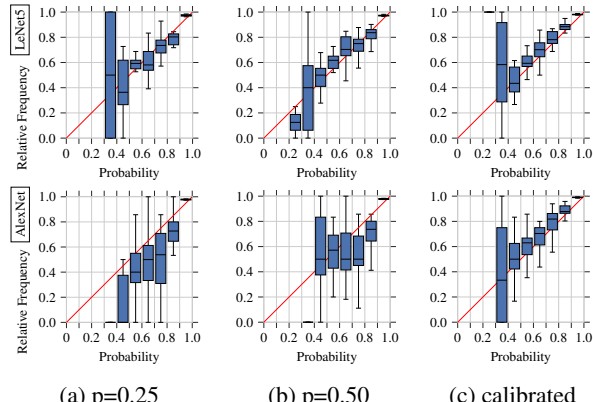

(a) p=0.25     (b) p=0.50     (c) calibrated

*Figure 2.* Reliability plots for calibration of frequentist models

The probabilistic behavior of a **variational** model allows the generation of multiple independent predictions for each input sample. The mean and median of the observed output distributions were calculated, and the calibration of these metrics was analyzed. In addition, the standard deviation was interpreted as a measure of uncertainty, as opposed to a confidence measure. Similar to the frequentist approach, the LeNet5 provided fairly well calibrated confidence scores for mean and median predictions. The reliability plots in Figure 3 present only larger deviations for lower prediction values, which can be explained by the smaller number of relevant predictions. Also the AlexNet presents a better initial calibration than in the frequentist setting but is still slightly overconfident. The standard deviation of variational predictions provides useful information. Unlike the confidence scores shown before, the standard deviation does not contribute to the decision making process of the model but is interpreted as uncertainty measure. While mean and median are moderately suitable for calibration, standard deviation and temperature scaling provided the best variational confidence optimization in terms of ECE and MCE for both models. As listed in Table 2, especially the MCE as a worst case scenario could be reduced, which is crucial for the underlying medical application.

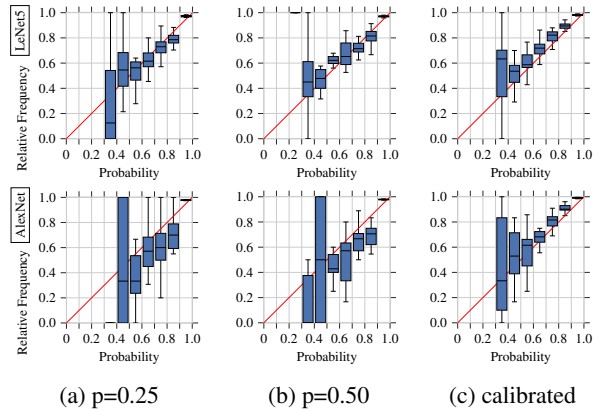

(a) p=0.25     (b) p=0.50     (c) calibrated

*Figure 3.* Reliability plots for calibration of variational models

|  | Dropout | VI | Metric | uncalibrated ECE | uncalibrated MCE | calibrated ECE | calibrated MCE | improvement ECE | improvement MCE |
|---|---|---|---|---|---|---|---|---|---|
| LeNet5 | p=0.00 | ✗ | — | 0.34 | 3.8 | **0.17** | 3.0 | 50.0% ↑ | 21.1% ↗ |
|  | p=0.25 | ✗ | — | 0.25 | 4.0 | 0.18 | 3.7 | 28.0% ↑ | 7.5% → |
|  | p=0.50 | ✗ | — | **0.21** | 3.0 | 0.20 | 2.8 | 4.8% → | 6.7% → |
|  | p=0.25 | ✓ | mean | 0.26 | 3.0 | 0.18 | 2.8 | 30.8% ↑ | 6.7% → |
|  | p=0.50 | ✓ | mean | 0.27 | 2.9 | 0.21 | 2.6 | 22.2% ↗ | 10.3% → |
|  | p=0.25 | ✓ | median | 0.26 | 3.1 | 0.18 | 2.9 | 30.8% ↑ | 6.5% → |
|  | p=0.50 | ✓ | median | 0.25 | 3.0 | 0.24 | 2.8 | 4.0% → | 6.7% → |
|  | p=0.25 | ✓ | std.dev. | **0.21** | **2.6** | 0.19 | 2.6 | 9.5% → | 0.0% → |
|  | p=0.50 | ✓ | std.dev. | 0.25 | 3.2 | 0.25 | **2.4** | 0.0% → | 25.0% ↑ |
| AlexNet | p=0.00 | ✗ | — | 0.26 | 4.9 | 0.22 | 4.0 | 15.4% ↗ | 18.4% ↗ |
|  | p=0.25 | ✗ | — | 0.26 | 4.9 | 0.18 | 3.8 | 30.8% ↑ | 22.4% ↗ |
|  | p=0.50 | ✗ | — | 0.26 | 4.0 | **0.12** | 3.4 | 53.8% ↑ | 15.0% ↗ |
|  | p=0.25 | ✓ | mean | 0.25 | 4.5 | 0.14 | 3.5 | 44.0% ↑ | 22.2% ↗ |
|  | p=0.50 | ✓ | mean | 0.25 | 4.7 | 0.14 | 3.5 | 44.0% ↑ | 25.5% ↑ |
|  | p=0.25 | ✓ | median | 0.25 | 4.0 | 0.14 | 3.7 | 44.0% ↑ | 7.5% → |
|  | p=0.50 | ✓ | median | 0.25 | 4.6 | 0.13 | 3.9 | 48.0% ↑ | 15.2% ↗ |
|  | p=0.25 | ✓ | std.dev. | 0.23 | **3.2** | 0.19 | **3.0** | 17.4% ↗ | 6.3% → |
|  | p=0.50 | ✓ | std.dev. | **0.18** | 3.4 | 0.15 | 3.1 | 16.7% ↗ | 8.8% → |

*Table 2.* Expected and maximum calibration error for all tested confidence measures averaged over 15 independent evaluation runs. Error values are stated in a magnitude of $10^{-1}$.

In summary, the results of examining frequentist and variational inference methods for LeNet5 and AlexNet architectures are consistent with the observation that confidence estimates from larger models tend to be miscalibrated (Guo et al., 2017). The smaller LeNet5 generated well-calibrated confidence estimates with considerably low and consistent deviations from ideal behavior. The more complex AlexNet architecture provided better classification results, but also produced overconfident predictions. Temperature scaling enabled the implementation of a large AlexNet with good classification performance and well-calibrated confidence estimates. Consulting the results of Table 1 and 2, the experiments showed that the calibrated AlexNet architecture with the dropout rate of $p = 0.50$ achieved the best $F_1$-scores and the lowest ECE values of all tested models.

### 4.3. Visual Explanations

As not all of the tested explanation approaches provided useful results for quantitative phase images, which are not as rich in features as macroscopic images, we will only provide the results for LIME and Guided Backpropagation. The analysis of Occlusions, Backpropagation and Grad-CAM are stated in the appendix in section A.4.

**LIME** explanations were not promising either, as their quality is highly dependent on the image segmentation approach used. Inspired by the principles of *tile coding* (Sherstov & Stone, 2005), best results were achieved by combining several sets of segmentations into one explanation. The interpretability was further improved by neglecting the original binary setting (Ribeiro et al., 2016a) and emphasizing the contributions of the individual areas according to their weight in the surrogate model. Figure 4 displays the results of the weighted outputs of LIME explanations on four superimposed SLIC segmentations (Further details on the optimization of the segmentation can be found in A.3). The

blue areas indicate a positive correspondence of the underlying cell structures and the predicted label, red areas show an opposing relation. Where the exemplary Monocyte and Lymphocyte exhibit a supporting explanation, larger red areas for the Neutrophil and Eosinophil examples might require a double check by a physician or biologist.

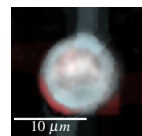 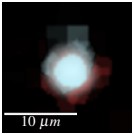 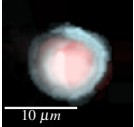 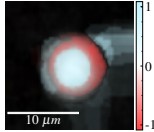

(a) Monocyte   (b) Lymphocyte   (c) Neutrophil   (d) Eosinophil

*Figure 4.* LIME explanations using SLIC-segmentation

**Guided Backpropagation** combines two approaches, by weighing the results of backpropagation with the class activation maps of Grad-CAM. Therefore, Guided Backpropagation cannot be applied to LeNet5. In most cases, the generated explanations in Figure 5 highlight only small parts of the cells, which could imply the detection of nucleus structures. For the samples of classes Lymphocyte and Eosinophil, the explanation also emphasizes minor gradients surrounding the actual cell, which could indicate that the size of the cell plays a role as other background parts in the distant corners are not affected.

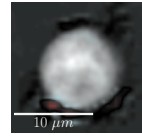 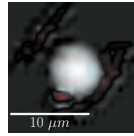 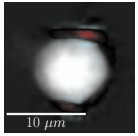 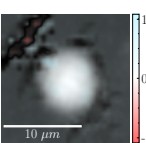

(a) Monocyte   (b) Lymphocyte   (c) Neutrophil   (d) Eosinophil

*Figure 5.* Explanations derived from Guided Backpropagation

### 4.4. Aggregated Meta-Explanations

With the huge number of cells to be analyzed, biomedical researchers only need the individual explanations in special cases. Usually, the general predictive behavior of the models is of greater interest. Therefore, in the following paragraphs, we will examine the models for general predictive patterns. One is based on ground truth labels and confidence scores, and the other is based on clustering methods. As LIME and Guided Backpropagation seemed to produce the most interpretable explanations, we will focus on those two approaches. To calculate the confidence estimates the variational scenario is used.

**Aggregation based on labels and confidence estimations**
For aggregating the individual explanations, the confidence estimates were grouped into six equally sized bins and separated by their class label. The resulting averaged meta-explanations for the calibrated LeNet5 can be seen in Figure 6. Especially for the most certain group, two distinct patters can be observed: Monocytes and Eosinophils are

← uncertain    certain →

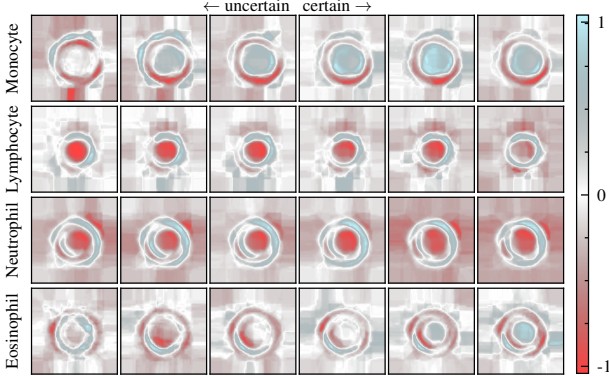

Figure 6. Aggregated LIME explanations based on calibrated confidence estimations for LeNet5

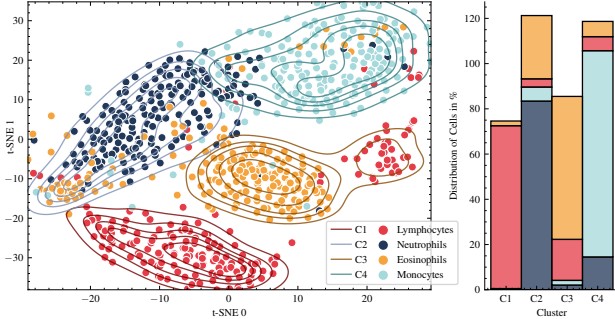

Figure 7. Clustering of t-SNE embedded LIME explanations from AlexNet by a k-Means algorithm. The clusters could not exactly assign all four classes. Therefore, the cumulative sum for some clusters is higher than 100% in the chart on the right.

represented by a stronger positive contribution of the inner part of the cells. In contrast, Neutrophils and Lymphocytes clearly depict a blue circle, which indicates the importance of the cell membrane. Furthermore, this behavior correlates with the biological appearance of the cells: The large Monocytes and small Lymphocytes can be easily differentiated from the other classes purely considering their size. For the more similar Neutrophils and Eosinophils, the network has learned to consider the cells' interior for one group to make a distinction.

In general, the prediction patterns for AlexNet and LeNet5 are similar. The LIME aggregations for AlexNet, displayed in Figure 19 (Appendix), entail an overall higher mean value, which makes the detected features less prominent. Also, the meta-explanations using guided backpropagation reveal a similar behavior. Figure 20 (Appendix) presents the same patterns for distinguishing the cells in their size as well as in their interior. For all classes the patterns get more precise with an increasing confidence estimate. Particularly, Eosinophils demonstrate the need for very confident estimates, to ensure that the correct parts of the image were analyzed for the clinical decision making.

**Aggregation based on explanation clustering** An alternative to the aggregation based on ground truth labels is the aggregation by unsupervised clustering. Here, we will see whether there are unique classification strategies that correspond to a particular cell type. In order to remain in a dimension that is manageable for humans, the high dimensional LIME explanations are embedded in a 2D space using t-SNE (van der Maaten & Hinton, 2008). This embedding is visualized in Figure 7, in which the color of the dots illustrates the respective cell class. Applying a k-Means clustering, with $k$ equal to the number of classes, on this 2D space reveals distinct detection patterns for each individual class. Solely cluster C3, which is dominated by Eosinophils, incorporates an apostate group of Lymphocytes. This might be due to a limited capture quality, reduced sample purity or the fact that the network uses two distinct strategies to

discover the small Lymphocytes. For the same reasons, also other clusters, especially C2, exhibit some mismatches as can be seen in the bar chart in Figure 7. Nevertheless, there is always one dominant cell class which supports our assumption that the network mainly relies on disjoint detection patterns for each of them.

## 5. Applications

After calibrating the confidence estimations and extracting patterns for the general predictive behavior of the networks, the presented techniques need to prove useful in real-world applications. Therefore, we confronted the variational setup for the LeNet5 with unknown data from familiar and unfamiliar domains and tested its classification confidence and the according visual explanations. We expect a high confidence only for leukocyte samples so the influence by unwanted objects stays at a minimum. In cases of overconfidence, the visual explanation should help to detect a violation of the general detection pattern in order to mitigate the interferences.

For an initial overview, we applied a train test split closer to real-world scenario to the leukocyte data. The test set of 1024 cells now consists exclusively of data from an independent donor, which was not present during training. To test resilience to typical error sources, we introduced two additional test sets: **Erythrocytes** make up 99% of human blood (Alberts, 2017), hence, it is likely that they find their way into leukocyte images. They should not be classified as leukocytes and need to exhibit a low confidence score. As the viscoelastic focusing by the microfluidics chip cannot guarantee perfect focus for all cells, **Defocused** examples should also be discarded by their low confidence score. In addition to erythrocytes and defocused cells, we picked two deviant test sets to simulate unfamiliar if not confusing inputs for the classifiers: The well known **MNIST** (LeCun et al., 2010) data set provides a similar image size but stands out with prominent edges. A data set of images with the

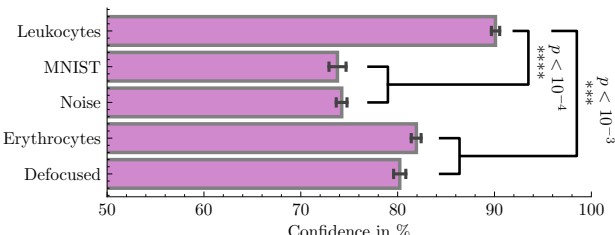

*Figure 8.* Confidence estimates by the LeNet5 for the different test sets. The error bars describe the standard error.

same dimensions but consisting purely of white **Noise** completes the list of challenges. The network could solve the leukocyte classification task for the new individual with an accuracy of 92,8% and a high confidence in its predictions, as Figure 8 displays. Additionally, the results demonstrate a general robustness against too deviant inputs. Calibrated on the standard deviation from variation inference, the confidence score shows a tremendous drop for MNIST and Noise images. Also, for the more closely related test sets, there is still a significant difference in the networks confidence, as determined by a *Kruskal-Wallis test* (Kruskal & Wallis, 1952) and post hoc analysis using *Bonferroni correction* (Armstrong, 2014).

### 5.1. Visual Inspection of Unknown Data

Even if the confidence estimation works well for most data and a clinical decision can be based exclusively on the most confident predictions, Figure 8 uncovers that there are still abnormal objects, which also reach a high confidence. Hence, Figure 9 investigates examples of unknown objects that could falsely contribute to the four-part differential. Here, the visual explanations of the noise patterns (a) and the MNIST image (b) show a totally divergent appearance which does not fit our general detection behavior. Some erythrocytes (c), nevertheless, could be too similar to leukocytes, as their outer cell membrane contributes positively to the prediction. Though, the inner torus shape should oppose a confident prediction.

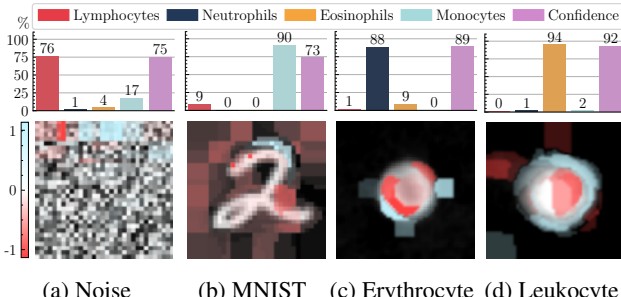

*Figure 9.* Examples of unknown data might have a relatively high confidence estimate but stand out by their visual explanations. The respective bar plots visualize the predicted class estimates and the according confidence score in %.

### 5.2. Outlier Detection by Visual Inspection

Moving on from unknown data, the network also has to deal with cells and structures which were present during training but should not influence the classification results as they are no valid leukocytes. For this purpose, Figure 10 displays some examples of those *outliers*, their predicted class, and the according explanation. These are thrombocytes (a), defocused cells (b) or ruptured cells (d). Micro-Thrombotic events, also called aggregates (c), might have their relevance for certain diseases (Nishikawa et al., 2021) but are inconvenient for the four-part differential. Thrombocytes and ruptured cells should not be a big problem, as they show a conflicting explanation pattern. However, the thrombocyte has a rather high confidence score, which could be problematic. Also the defocused cell gets recognized, which is rather exceptional. The biggest problem still are aggregates as they contain more than one cell. The explanation in Figure 10c therefore has two contribution regions resulting in a high confidence, but two cells of different types would cancel each other out. Consequently, the proposed method offers only limited help for outlier detection, but the concerned objects are easily detectable using other methods. Stricter filter rules or more advanced techniques for this use case as presented by Röhrl et al. (2022) are strongly recommended.

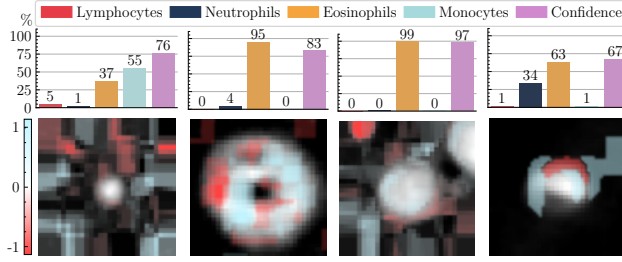

(a) Thrombocyte  (b) Defocused  (c) Aggregate  (d) Ruptured

*Figure 10.* Some kinds of outliers from the leukocyte data set might be difficult to detect as they show a high confidence and partly similar explanation patterns. The respective bar plots visualize the predicted class estimates and the according confidence score in %.

### 5.3. Mislabeled Data

Finally, there were some discrepancies during the training of the networks. Normally, we would expect the classification error to shrink with an increasing confidence, but for certain samples this was not the case. This is based on the fact that hardly any biological sample is of 100% purity. For the creation of the four-part differential training data set, this originates from the separation process of the individual cell types via *immunomagnetic isolation kits*. Here, paramagnetic antibodies are used to label and sort the cells. It might happen that some of the cells escape this labeling and contaminate the other classes. Modern isolation kits reach a purity of 95% and above (Son et al., 2017) and are constantly improved.

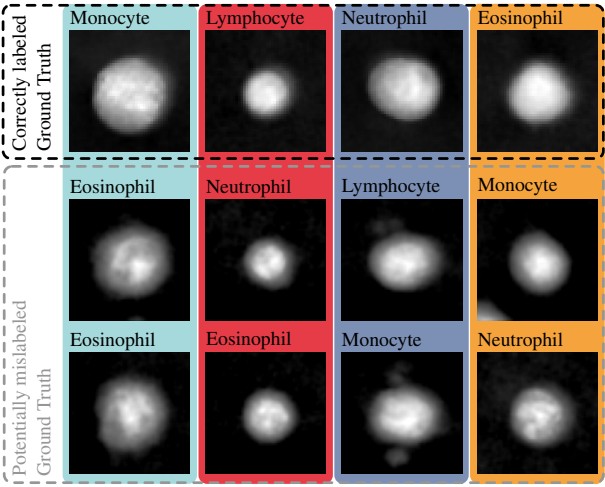

*Figure 11.* The top row shows valid representatives of the ground truth label. The lower rows contain potentially mislabeled cells that were assigned to another class by the LeNet5 with a confidence estimate ≥ 95%.

Nevertheless, our calibrated networks could demonstrate that they detected cells with a high confidence estimate for a potentially wrong class. Inspecting these images showed that the network might have become smarter than the ground truth, as Figure 11 reveals. Obviously, the cell in the top row has more resemblance to the cells drawn in the same column below than to the cell class the ground truth label would imply. Hence, the proposed method could be used in a *human-assisted labeling* setting (Holzinger, 2016) to further purify biological data sets.

## 6. Discussion and Conclusion

The goal of this work was to improve the interpretability of machine learning to overcome the limited applicability of algorithmic decision making in a clinical environment. For this purpose, we chose the ascending and label-free platform technology of QPI and performed a four-part differential of leukocytes, as there are many publications for the proof of concept but none which focus on transparency.

For the selected use case, the vanilla AlexNet model showed slightly better classification performance than the smaller LeNet5, but with a higher overestimation of its confidence. This drawback was overcome by introducing temperature scaling as an effective way to **calibrate** the confidence estimations. The application of variational inference further improved the consistency of the confidence estimation and reduced the ECE and MCE. Together with the high classification accuracy, these values can be used as a quality measure in a certification processes, when the presented techniques are integrated in a medical assay (Jin et al., 2023).

The comparison of state-of-the-art **visualization methods** for deep learning predictions outlined promising results for LIME and Guided Backpropagation. The methods fa-

cilitated the visualization of relevant decision factors for individual predictions. LIME was further adapted to convey the relative importance of individual image regions and to achieve explanations with higher granularity. It was possible to derive consistent meta-explanations and extract general detection patterns by aggregation or unsupervised clustering. Nevertheless, the appearing patterns had only limited resemblance with the **biological patterns** we hoped to find. As Figure 6 outlines, Monocytes and Lymphocytes are differentiated by their unique size. Eosinophils and Neutrophils generally have a similar appearance, but the networks are able to tell them apart based on their interior which seems to be more emphasized in the case of Eosinophils.

Applying the optimized technology to unknown data in **real-world scenarios** revealed high robustness against deviant cell structures and contamination. Certainly, leukocytes from new donors were accurately classified with a high degree of confidence. The calibrated confidence estimation even allowed the detection of mislabeled cells in the ground truth. For outlier detection like thrombocytes, aggregates, defocused or ruptured cells, the methods did not perform well and we recommend to use other methods for this purpose.

The high and robust classification performance without special labeling procedures demonstrates the maturity of the technologies presented in this and related work (Ugele et al., 2018; Shu et al., 2021; Fanous et al., 2022). However, if the essential explainability of the decisions is still missing and there is no relation to the established biological features, the clinical acceptance and a market entry will remain challenging. Therefore, in **future work** we want to use newer methods for visual explanations, such as *Smooth Grad-CAM++* (Omeiza et al., 2019), *EVET* (Oh et al., 2021), or *FIMF score-CAM* (Li et al., 2023) to help QPI make the expected breakthrough (Nguyen et al., 2022). We hope that others will also decide to integrate visual explanation and confidence calibration instead of focusing only on accuracy, in order to promote the discourse and enable interdisciplinary work on this topic (Yang et al., 2020).

All in all, the work contributes to making deep learning more transparent and communicable for the investigated use case. It represents a development in the right direction of **interpretable** machine learning in this field and lays the foundation for subsequent user studies in biomedical research and clinical application.

## Acknowledgements

The authors would like to especially honor the contributions of J. Groll for the software implementation and experiments. This research was funded by the German Federal Ministry for Education and Research (BMBF) with the funding ID ZN 01 | S17049.

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

# A. Appendix

## A.1. Image Preprocessing

To achieve satisfactory segmentation and classification results, it is crucial to perform preprocessing. Figure 12a shows that the QPI configuration produces phase images of numerous cells with dimensions of 512×382 pixels. From this, we need to extract patches of 50×50 pixels containing only single cells in order to classify them properly.

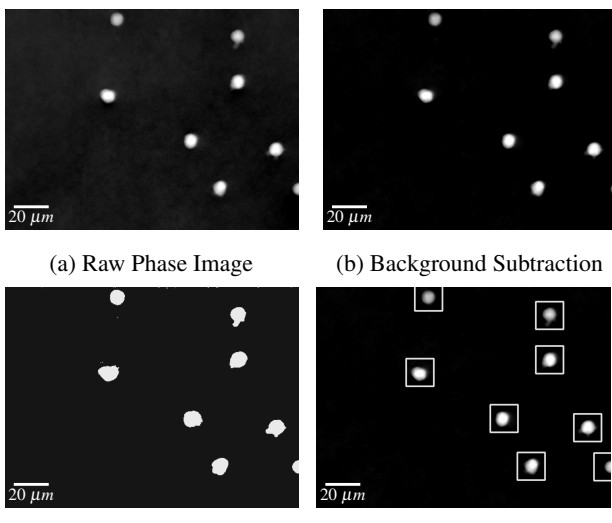

(a) Raw Phase Image      (b) Background Subtraction

(c) Threshold Segmentation      (d) Cell Image Patches

*Figure 12.* Preprocessing steps to achieve single cell image patches from a raw phase image

**Background Subtraction** To eliminate unwanted artifacts and background noise, the median of 100 images is computed and then subtracted from each frame. The transition from Figure 12a to Figure 12b visualizes the achieved smoothness in the image background. This operation is possible since the imaging setup and microfulidics channel are regarded as stationary.

**Segmentation** Cell detection in the acquired images involves two steps: thresholding and contour finding. First, the phase images are subjected to binary thresholding. Next, contours are extracted from the binary images using the algorithm introduced by Suzuki & Abe (1985). These extracted contours are then subjected to filtering based on a minimum contour area. Finally, each cell represented by a contour is saved as an image patch with dimensions of 50×50 pixels for further analysis. Compare Figures 12c and 12d.

**Normalization** Normalization is an essential step in preparing data for machine learning, especially when using neural networks, as it standardizes the feature or image values to a uniform range. The most common techniques are either mean and standard deviation based (such as z-score normalization) or minimum-maximum based (Singh & Singh, 2020; Kotsiantis et al., 2007). In this work, the images were first clipped to limit the range of values, since images produced by holographic microscopes theoretically have an unlimited range of values. Specifically, a minimum clipping value of 0.2 (due to the background) and a maximum clipping value of 4 were used, which proved effective in capturing leukocytes while minimizing cell clipping and utilizing the entire value range. *Min-Max normalization* was then used to transform the image values to the interval $[0, 1]$, which is ideal for neural networks.

## A.2. Morphological Features

Hand crafted features are widely spread in the cytology community. Therefore, we adapted their use in our work to perform some kind of quality control. Table 3 shows a subset of the features introduced by Kasprowicz et al. (2017), Ugele et al. (2018), and Paidi et al. (2021) which are sufficient to filter out artifacts and impurities of the blood samples. These features are manly based on OpenCV

| | Feature | Explanation | Unit |
|---|---|---|---|
| $P$ | # pixels | Number of pixels per cell contour | - |
| $\phi_i$ | phase shift | measured phase shift of the $i$-th pixel | rad |
| $\lambda$ | wave length | wave length of the light source (528nm) | nm |
| $A$ | area | $P \cdot$ pixel area | µm² |
| $d$ | diameter | $\sqrt{\frac{4A}{\pi}}$ | µm |
| $V$ | optical volume | $\sum^{P} \phi_i \cdot \frac{\lambda}{2\pi}$ | µm³ |
| $L$ | perimeter | OpenCV `arcLength()` of the cell contour | µm |
| $C$ | circularity | $\frac{4\pi \cdot A}{L^2}$ | - |

*Table 3.* Morphological Features (excerpt) adapted from (Kasprowicz et al., 2017; Ugele et al., 2018; Paidi et al., 2021)

contours[1] and the contained pixel values. They typically look like the contour drawn in red on the cell image in Figure 13. As texture features have proven to be insufficient for robust cell classification we do not consider them in this work (Röhrl et al., 2020).

---

[1] https://docs.opencv.org/3.4/dd/d49/ tutorial_py_contour_features.html

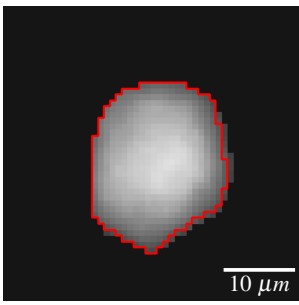

*Figure 13.* Cell image with its detected contour

## A.3. LIME Segmentation

The generation of meaningful LIME explanations requires an interpretable data representation. In a first step, different algorithms were implemented and compared to calculate a consistent segmentation of the cell images. To evaluate these, segmentation results for individual samples of all relevant classes of leukocytes were manually reviewed. The evaluation revealed that all tested algorithms require careful tuning of the respective parameters to achieve satisfactory outputs for all relevant cell types. Due to the high contrast between the actual cell and the background of an image, reasonable segmentation had to be ensured to differentiate the individual parts within a cell. This was necessary to obtain granular explanations that take into account both the background of an image and the internal structure of the captured cells. (Compare Figure 14)

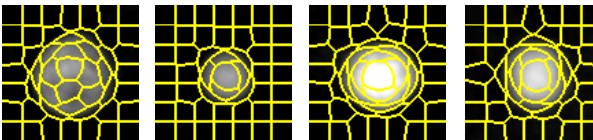

(a) Monocyte  (b) Lymphocyte  (c) Neutrophil  (d) Eosinophil

*Figure 14.* SLIC segmentation of cell images as a pre-processing step for LIME explanations

In order to enable LIME to also evaluate more granular regions, we tested the options to increase the number of segments per explanation or to combine the outputs from several segmentations for the same sample image. The first approach, to simply increase the number of segments and thus yielding a more detailed resolution, resulted in noisy and complex interpretations, which were counterintuitive. Combining and weighing several segmentations with different but constant numbers of segments was promising as can be seen in Figure 15. The final segmenter consists of $S=4$ individually configured SLIC approaches. Detailed settings for the SLIC segmentations can be found in Table 4. The weighted results of the according LIME explanations are then merged into one mask via averaging.

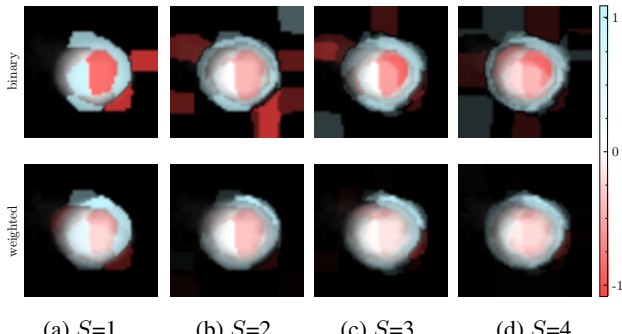

(a) $S=1$    (b) $S=2$    (c) $S=3$    (d) $S=4$

*Figure 15.* Effects of varying the number of $S$ SLIC segmentations on the weighted and binary LIME explanations

| Name | Segments | Compactness | Sigma |
|------|----------|-------------|-------|
| SLIC$_1$ | 15 | 10 | 3.0 |
| SLIC$_2$ | 25 | 10 | 2.5 |
| SLIC$_3$ | 35 | 25 | 3.0 |
| SLIC$_4$ | 50 | 15 | 5.0 |

*Table 4.* Parameterization of the used SLIC segmentations

## A.4. Visual Explanation

The visual explanation methods in this section were implemented and tested on the leukocyte data set presented. Unfortunately, they did not prove to be very helpful for our use case, but we would still like to show the results for comparison.

**Perturbation-based** approaches provide explanations by analyzing the effects of local changes on a model's response. These can also be model-agnostic as in case of simple occlusions (Zeiler & Fergus, 2014). Here, different image areas are systematically covered to determine the influence of the respective feature. To also detect cross-relationships between different areas, model-specific gradient information needs to be considered (Simonyan et al., 2014; Ancona et al., 2017). So called *meaningful perturbation* was introduced by Fong & Vedaldi (2017) to achieve more natural and plausible imaging. Instead of covering individual areas of an image with a black square, random noise and blur are applied to erase information in these specific areas. In the following example, simple occlusion was used with a patch of the size of 6×6 pixels to iteratively cover certain parts of an input image of 50×50 pixels. By observing the resulting changes in the prediction values, we calculate a sensitivity value for each pixel as shown in Figure 16. While the explanations roughly highlight relevant areas of an image, it is difficult to correlate the results with the underlying cells. Additionally, we noticed a high impact of the chosen patch size on the resulting sensitivity values, leading to inconsistent results.

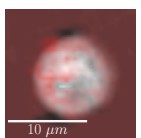 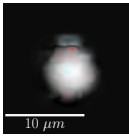 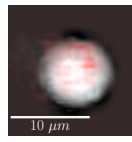 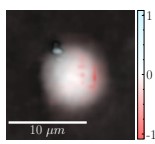

(a) Monocyte  (b) Lymphocyte  (c) Neutrophil  (d) Eosinophil

*Figure 16.* Explanations derived from Occlusion

**Backpropagation** uses the inner structure of the analyzed deep learning model to pipe back the prediction value to the initial input space. The resulting explanations give an indication of which patterns in the cell image triggered the activation of the neural networks. Therefore, the explanations presented are highly dependent on the actual size of the input space. The explanations for an AlexNet model, displayed in Figure 17, have a higher resolution compared to a LeNet5 model and are thus easier to interpret. Although the interpretation of these patterns is not obvious, certain parts can be attributed to either an internal structure of a cell or the high contrast of the outer membrane.

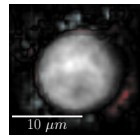 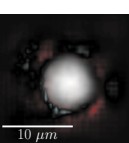 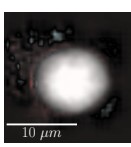 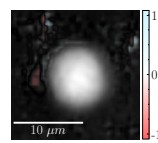

(a) Monocyte  (b) Lymphocyte  (c) Neutrophil  (d) Eosinophil

*Figure 17.* Explanations derived from Backpropagation

**Grad-CAM** explanation focuses on important regions of the image and produces much smoother results than the previously shown Backpropagation approach. However, this technique requires that the dimension of the last convolution block of the model is multidimensional, thus preventing its application to the LeNet5. The final convolutional layer of the implemented AlexNet architecture consists of filters with a size of 13×13. The total activation of this filter was aggregated and interpolated to fit the original, higher-dimensional input space. Therefore, the class activation maps had a low resolution, which directly depended on the underlying model architecture. As shown in Figure 18, Grad-CAM can be used as a basic method to validate relevant domains for a model but at the same time, the information is limited and does not allow for further differentiation.

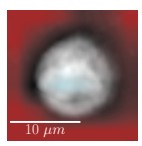 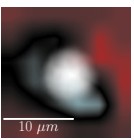 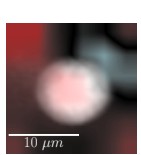 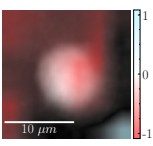

(a) Monocyte  (b) Lymphocyte  (c) Neutrophil  (d) Eosinophil

*Figure 18.* Explanations derived from Grad-CAM

## A.5. Aggregation

For the AlexNet architecture it was also possible to extract general detection patterns for the different leukocyte classes. The pattern for LIME does not change that much as plotted in Figure 19. On the other hand, Guided Backpropagation produces clearly evolving patterns with an increasing confidence, which can be seen in Figure 20. First, the detection pattern focuses much more on the background, whereas for a higher confidence score, the attention moves towards the actual cells.

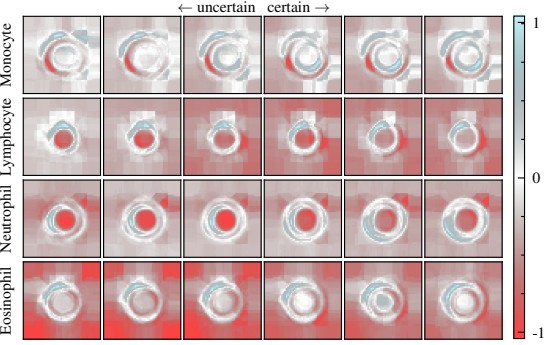

*Figure 19.* Aggregated LIME explanations based on calibrated confidence estimations for AlexNet

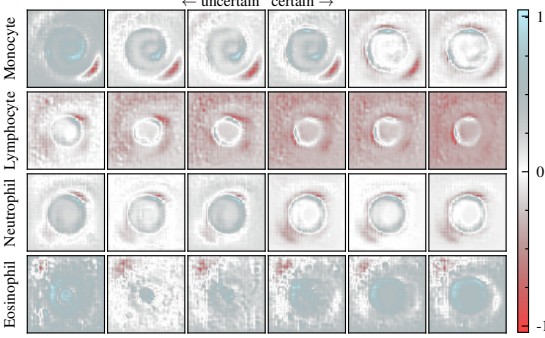

*Figure 20.* Aggregated Guided Backpropagation explanations based on calibrated confidence estimations for AlexNet

