# OpenReview forum: "Towards Interpretable Classification of Leukocytes based on Deep Learning"
_ICML.cc/2023/Workshop/IMLH — IMLH 2023 Poster_

### Official Review · Reviewer_vvdE · 2023-06-07
**A new way for visual based explanation**

**Rating:** 7
**Confidence:** 4

**Review:**

This paper proposed a new way to visualize the explanations in cytology images. It aggregates the LIME results for a general explanation which is more useful for pathologists.

---

### Official Review · Reviewer_4Wxd · 2023-06-18
**Interesting work for leukocytes classification**

**Rating:** 7
**Confidence:** 3

**Review:**

This work investigates the calibration of confidence estimation for the automated classification of leukocytes and provides interpretable visual explanations for the proposed model. This topic is very interesting and the quantitative and qualitative results both look strong.

---

### Official Review · Reviewer_peuy · 2023-06-19
**The paper implements AlexNet and LeNet5 to the classification of leukocytes**

**Rating:** 6
**Confidence:** 2

**Review:**

The paper implements AlexNet and LeNet5 to the classification of leukocytes using QPI. It carries out a comprehensive analysis on differentiating leukocyte subtypes, modifications for confidence estimation, visual explanation tools, and meta-aggregations to derive general detection patterns.

Overall, I didn't see substantial methodological innovations in this study. I may not be able to provide a good assessment about the importance and significance of this paper as a benchmark study of existing models, due to my lack of expertise in the field of leukocyte imaging.

---

### Meta-Review · Area_Chair_NqA6 · 2023-06-20

**Recommendation:** Accept (Poster)
**Confidence:** 5

**Metareview:**

This paper implements AlexNet and LeNet5 for leukocyte classification using QPI and explores various aspects including subtype differentiation, confidence estimation modifications, and visual explanations. Reviewers find the work interesting, appreciate the strong quantitative and qualitative results, rating the paper as good and acceptable.

---

### Decision · Program_Chairs · 2023-06-20

Accept (Poster)